# Integrated model for food-energy-water (FEW) nexus to study global sustainability: The main generalized global sustainability model (GGSM)

**Apoorva Nisal**[1☯], **Urmila Diwekar**[1,2☯]\*, **Neeraj Hanumante**[3‡], **Yogendra Shastri**[3‡], **Heriberto Cabezas**[4‡]

**1** Department of Industrial Engineering, University of Illinois, Chicago, IL, United States of America, **2** Vishwamitra Research Institute, Crystal Lake, IL, United States of America, **3** Department of Chemical Engineering, Indian Institute of Technology Bombay, Mumbai, Maharashtra, India, **4** Research Institute for Applied Earth Sciences, University of Miskolc, Miskolc, Hungary

☯ These authors contributed equally to this work.
‡ NH, YS and HC also contributed equally to this work.
\* urmila@vri-custom.org

**Data Availability Statement:** All the data is available from: https://zenodo.org/record/6331602.

## Abstract

Over the years, several global models have been proposed to forecast global sustainability, provide a framework for sustainable policy-making, or to study sustainability across the FEW nexus. An integrated model is presented here with components like food-web ecosystem dynamics, microeconomics components, including energy producers and industries, and various socio-techno-economic policy dimensions. The model consists of 15 compartments representing a simplified ecological food-web set in a macroeconomic framework along with a rudimentary legal system. The food-web is modeled by Lotka–Volterra type expressions, whereas the economy is represented by a price-setting model wherein firms and human households attempt to maximize their economic well-being. The model development is done using global-scale data for stocks and flows of food, energy, and water, which were used to parameterize this model. Appropriate proportions for some of the ecological compartments like herbivores and carnivores are used to model those compartments. The modeling of the human compartment was carried out using historical data for the global mortality rate. Historical data were used to parameterize the model. Data for key variables like the human population, GDP growth, greenhouse gases like $CO_2$ and $NO_X$ emissions were used to validate the model. The model was then used to make long-term forecasts and to study global sustainability over an extended time. The purpose of this study was to create a global model which can provide techno-socio-economic policy solutions for global sustainability. Further, scenario analysis was conducted for cases where the human population or human consumption increases rapidly to observe the impact on the sustainability of the planet over the next century. The results indicated that the planet can support increased population if the per capita consumption levels do not rise. However, increased consumption resulted in exhaustion of natural resources and increased the $CO_2$ emissions by a multiple of 100.

**Funding:** NH, YS, UD acknowledge the support from the Ministry of Human Resource Development, Government of India, through the SPARC (project code: P1238). The research contribution by H. Cabezas was carried out in the GINOP-2.3.2-15-2016-00010 framework Development of enhanced engineering methods with the aim at utilization of subterranean energy resources" project at the Research Institute of Applied Earth Sciences of the University of Miskolc, the Széchenyi 2020 Plan, partially funded by the European Union, co-financed by the European Structural and Investment Funds. The funders had no role in study design, data collection and analysis, decision to publish, or preparation of the manuscript.

**Competing interests:** The authors have declared that no competing interests exist.

**Abbreviations:** $g_{ESIRP}$, Mass transfer from Fuel Source to Inaccessible Resource Pool; $g_{H1C1}$, Mass transfer from $H_1$ to $C_1$; $g_{H1HH}$, Mass transfer from $H_1$ to HH; $g_{H2C1}$, Mass transfer from $H_2$ to $C_1$; $g_{H2C2}$, Mass transfer from $H_2$ to $C_2$; $g_{H3C2}$, Mass transfer from $H_3$ to $C_2$; $g_{IRPRP}$, Mass transfer from Inaccessible Resource Pool to Resource Pool; $g_{ISIRP}$, Mass transfer from Industry Sector to Inaccessible Resource Pool; $g_{P1H1}$, Mass transfer from $P_1$ to $H_1$; $g_{P1H2}$, Mass transfer from $P_1$ to $H_2$; $g_{P1HH}$, Mass transfer from $P_1$ to HH; $g_{P1IS}$, Mass transfer from $P_1$ to IS; $g_{P2H1}$, Mass transfer from $P_2$ to $H_1$; $g_{P2H2}$, Mass transfer from $P_2$ to $H_2$; $g_{P2H3}$, Mass transfer from $P_2$ to $H_3$; $g_{P3H3}$, Mass transfer from $P_3$ to $H_3$; $g_{RPIRP}$, Mass transfer from Resource Pool to Inaccessible Resource Pool; $g_{RPIS}$, Mass transfer from Resource Pool to Industry Sector; $g_{RPP1}$, Mass transfer from resource pool to $P_1$; $g_{RPP2}$, Mass transfer from resource pool to $P_2$; $g_{RPP3}$, Mass transfer from resource pool to $P_3$; $m_{C1}$, Mortality rate of $C_1$; $m_{C2}$, Mortality rate of $C_2$; $m_{H1}$, Mortality rate of $H_1$; $m_{H2}$, Mortality rate of $H_2$; $m_{H3}$, Mortality rate of $H_3$; $m_{HH}$, Mortality rate of humans; $m_{P1}$, Mortality rate of $P_1$; $m_{P2}$, Mortality rate of $P_2$; $m_{P3}$, Mortality rate of $P_3$; $r_{IRPP2}$, Recycling term for mass transfer from Inaccessible Resource Pool to $P_2$; $r_{IRPP3}$, Recycling term for mass transfer from Inaccessible Resource Pool to $P_3$; $y_{C1}$, Mass of carnivore $C_1$; $y_{C2}$, Mass of carnivore $C_2$; $y_{H1}$, Mass of herbivore $H_1$; $y_{H2}$, Mass of herbivore $H_2$; $y_{H3}$, Mass of herbivore $H_3$; $y_{HH}$, Mass of Human Households (HH); $y_{IRP}$, Mass of Inaccessible Resource Pool (IRP); $y_{IS}$, Mass of the Industry Sector (IS); $y_{P1}$, Mass of primary producer $P_1$; $y_{P2}$, Mass of primary producer $P_2$; $y_{P3}$, Mass of primary producer $P_3$; $y_{RP}$, Mass of Resource Pool (RP).

## Introduction

The Brundtland commission report [1] defines sustainability as, "development that meets the needs of the present without compromising the ability of the future generations to meet their own needs." While different societies have a varied understanding of sustainability, Brown and team propose a crude definition for sustainability as the basic structure of support systems that can be maintained only with a healthy environment and a stable human population which is necessary for indefinite human survival on a global scale [2]. They state that the key elements of sustainable development include continued support of human life on Earth, long-term maintenance of the stock of biological resources and the productivity of agricultural systems, stable human population, environmentally feasible economic growth, an emphasis on smaller-scale and self-reliance, and continued quality in the environment and ecosystems. Thus, sustainable development can only be achieved by addressing several interdependent ecological and socio-economic systems and the complex relationships within these systems.

An increase in population, global economic growth, and technological and scientific advancement have created their share of issues such as increased pollution, ecosystem degradation, and extinction of species and resources. Dietz(2007) showed that increased affluence combined with population growth exacerbates the impact on the environment and substantially increases the human footprint on the planet [3]. They showed that focused international efforts, increased technological efficiency, and strategic policies are necessary to minimize the environmental impact and reduce human stress on the environment. There has been a worldwide influx in strategic policy-making, with sustainable development being the goal. In 2016, the United Nations proposed 17 sustainable development goals focusing on food, land, energy, water, climate, infrastructure, socio-economic growth, and equality [4]. Global development through the last century has led to growth in the economic activity while also affecting natural resources and the environment. There is a need for global policy making to provide guidelines for sustainable development and economic growth. Global models depicting the state of the world help in the study of the complex interconnected dynamics of sustainable growth and provide a scientific basis for the decision-making process behind global policies.

The World 3 model was developed to gain insights into the limits of the world system and the constraints it put on human numbers and activity. It also aimed to help identify and study the dominant elements and their interactions that influence the long-term behaviour of world systems [5]. It is a system dynamics model with target sectors for food production, industry, population, pollution, and non-renewable resources [6]. World 3 model was the first of its kind global model, which paved the path for future global models. Following the World 3 model, Global Unified Model of the BiOsphere (GUMBO) was another global model which followed the ideology of creating a metamodel of the "biosphere", deriving inspiration from geological, global climate, sociology, economic, atmospheric, and ecosystem models [7]. The inputs to the GUMBO model are the outputs generated by more complex, computationally intense models, and its calibrations rely on existing observational databases. GUMBO improved upon the World3 model by adding relative rates of investment which can be modified by the user to observe resulting impacts on the model output.

A world model was also developed by Puliafito and team, using coupled differential equations to describe the changes of population and economic growth or gross domestic product (GDP) modeled as competing species as in Lotka–Volterra prey–predator relations [8]. They forecasted the evolution of the world population, gross domestic product (GDP), primary energy consumption, and carbon emissions from the year 1850 to year 2150. Motesharrei and team developed a Human and Nature Dynamics (HANDY) Model based on predator-prey approach for humans and nature as competing species [9]. They added accumulated wealth

and economic inequality along with a human population dynamics model. Their simulated scenarios offer significant implications which describe the socio-economic strata and effects on sustainability in the form of social well-being. Another conceptual world Earth model was proposed to qualitatively represent the global co-evolutionary dynamics of humans and nature for different socio-cultural stages of human history on Earth and study the mid to long-term evolution of Earth [10]. An Earth model (Earth$_3$) aimed at determining the feasibility of achieving the 17 Sustainable Development Goals (SDG) proposed by the United Nations (UN) till 2030 [11] has also been developed recently. This model combined a socio-economic model of human activity with a biophysical model of the global environment. Their predictions show that in the current state regime, the social and environmental SDGs cannot be achieved together. King(2020) created a dynamic growth model inspired from the HANDY model linking biophysical and economic variables or Humans and Resources with money (HARMO-NEY), which enabled them to study inter-linkages between resource extraction and depletion, population, capital, debt, and money flow within the economy [12]. Their results indicated that while higher investment in capital extracted resources more quickly, it also prevented a full rapid collapse as population grew at a lesser rate. Their model framework was created to further add features and to calibrate real-world economies for exploration of future energy-economic scenarios, such as transition to low-carbon energy. HANDY, Earth-3, and HARMO-NEY models are all global models primarily concentrating on sustainable economic development.

Recently, Van Vuuren and team analysed the food-energy-water nexus through integrated scenarios and verified the impact of several techno-socio-economic policies on global sustainability throughout 2015-2020 [13]. Their model studied the impact of development on agriculture, energy, water, land use and greenhouse gas emission utilizing the IMAGE 3.0 model. The generalized global sustainability model (GGSM) presented here is targeted at multiple trophic levels with intra-trophic level diversity and includes a market mechanism along with choice of industry for employment which sets it apart from the work by Van Vuuren et al. A model based on an integrated economy under imperfect competition combined with a twelve-cell ecological model was previously developed [14, 15]. The twelve-cell model mimicked a global ecosystem with a rudimentary social system that regulated the flows of mass according to its criteria. In the next phase, an industrial sector and the rudimentary economic system were added to this model with ecology, economics, social system, technology, and time being modeled together [16, 17]. The food web model was created using Lotka–Volterra type predator-prey expressions, whereas the economy was represented by a price-setting model wherein firms and human households attempted to maximize their well-being. All relevant global models forecast scenarios and the impact on resources; however, none of them have the capability to provide techno-socio-economic policies to make sustainable changes owing to their descriptive nature. In contrast, the generalized global sustainability model (GGSM) presented here is developed to be descriptive as well as prescriptive. It thus aims at providing guidance on feasible techno-socio-economic policies to make strides towards sustainable growth. GGSM focuses on global sector wide effects, gross domestic product (GDP) and greenhouse gas emissions such as carbon dioxide and nitrous oxide emissions. Thus, it adopts a granular approach by nature.

In this work, the proposed food-web model is combined with a microeconomics model similar to the one proposed by Kotecha(2013) [17]. In the work by Kotecha (2013), the baseline population was assumed to be constant. The current model addresses this limitation and improves on the previous work by adding predictors for gross domestic product and greenhouse gas emissions which makes it better suited to address important global issues. GGSM is then used to analyse forecasted scenarios for population explosion and increased consumption

on the sustainability of the planet for the next century. The impact and feasibility of achieving UN sustainable development goals in the domain of food, energy, and water, and inter-linkages that restrict achieving the targets are studied. Another aspect which differentiates the new 15 compartment model from other global models is the ability to provide strategic policy guidelines and evaluation of solutions to ensure long-term sustainability. This paper presents a validated model based on historical data and evaluation of several sustainability scenarios based on this validated model. Reparameterization of the model using historical data and validating the model results with the historical trends is one of the important contributions of this work. This model can prescribe techno-socio-economic policies, but the evaluation of such policies is out of the scope for the current study. The validated model sets the foundation for using the model for policy analysis through an optimal control theory-based approach. The detailed description of the generalized global sustainability model (GGSM) is presented in section 2 of this article. The parameterization and validation of the GGSM approach is presented in section 3 and the analysis of different sustainability scenarios with GGSM is presented separately in section 4 followed by the conclusions in section 5.

## Generalized Global Sustainability Model (GGSM)

In this section, the generalized global sustainability model and its various features will be described. The current model is conceptually similar to the previous model [17], which itself is an evolutionary development of an earlier model [14, 15]. The exception of this work is it being a global model and focusing on the food-energy-water (FEW) nexus. GGSM was created to represent a simplified global ecosystem with enough detail to allow the pursuit of further study. The 15-compartment model consists of a simplified ecological food web set in a macro-economic framework and a rudimentary legal system [17]. The different compartments in the model, shown in Fig 1, are three primary producers ($P_1$, $P_2$, and $P_3$), three herbivores ($H_1$, $H_2$, and $H_3$), two carnivores ($C_1$ and $C_2$), human households (HH), industry (IS), energy producer (EP), fuel source, water reservoir, resource pool (RP) and an inaccessible resource pool (IRP). The resource pool represents a finite nutrient source, while the inaccessible resource pool represents mass that is not biologically accessible to the rest of the ecosystem, perhaps due to pollution or other reasons like waste plastic. The resource pool is consumed by the primary producers to make mass biologically available for the rest of the ecosystem. The primary producers $P_2$ and $P_3$ recycle a small amount of mass from the inaccessible resource pool back into the system to represent bacterial actions. The mortality of all the nine biological compartments causes the mass to be recycled back to RP.

The fuel source is a finite non-renewable energy source. The ecological compartments cannot generate usable energy directly from the energy source as the fuel source needs to be appropriately transformed into power. The energy producer (EP) represents an industry that uses labor to transform the fuel source into usable energy. Subsequently, this energy is then utilized by the human households (HH) and the industrial sector (IS). It is assumed that the EP also has the capability to produce energy using $P_1$ to represent the biopower or the energy derived from biomass in this study.

The IS uses $P_1$ and the resource pool to produce products valuable to humans (HH). The mass of the HH does not increase by utilizing the IS products; however, the mass of the inaccessible resource pool does increase. Similarly, the mass of the inaccessible resource pool increases through energy production by EP by consuming mass and a consequent decrease in the mass of the fuel source. The biological compartments of the systems can be grouped into species with economic value or *domesticated species* and those with no economic value or *wild species*. A rudimentary legal system assigns property rights to the owners of domesticated

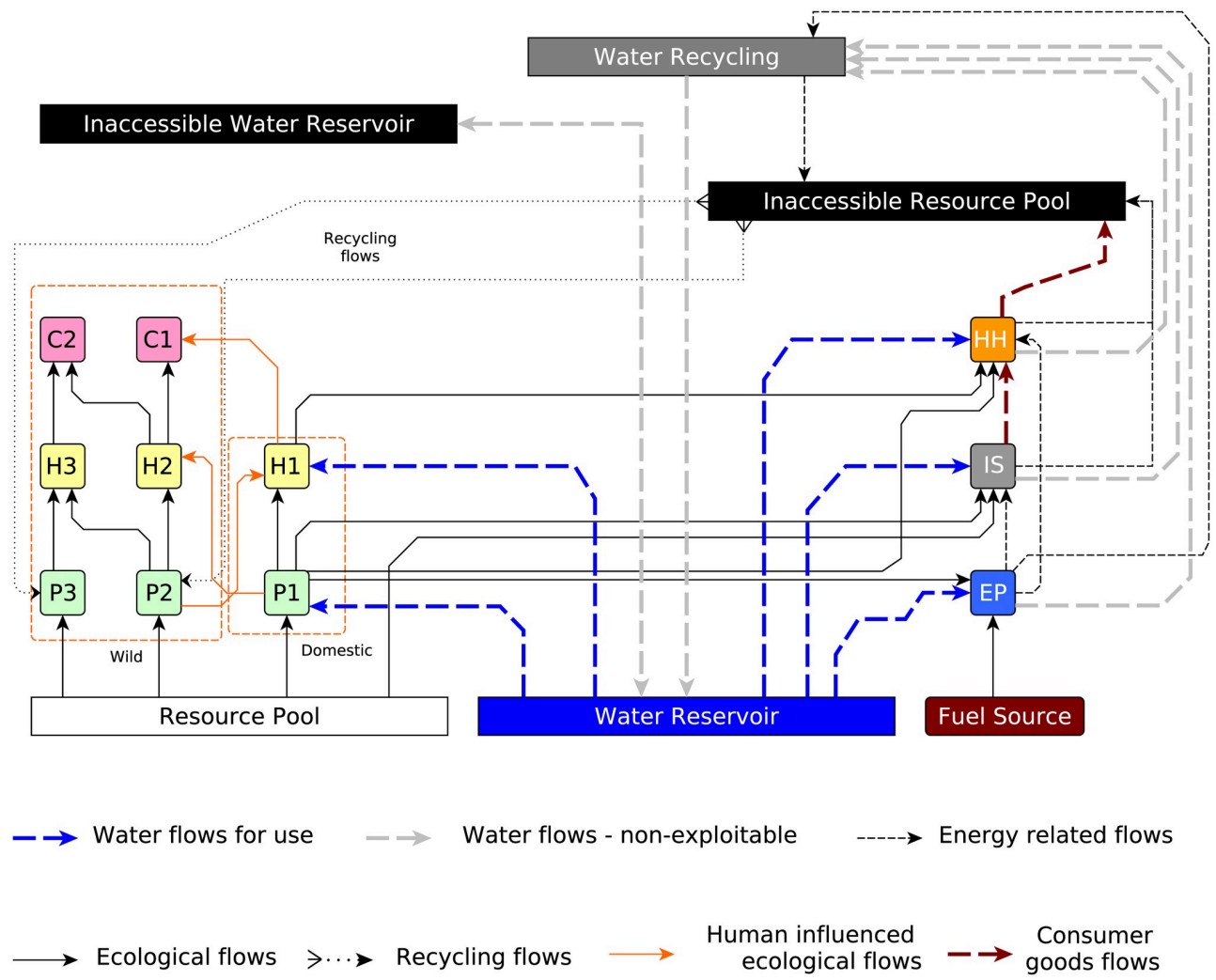

**Fig 1. Generalized Global Sustainability Model (GGSM).** Compartments: primary producers ($P_1$,$P_2$,$P_3$), herbivores ($H_1$,$H_2$,$H_3$), carnivores ($C_1$, $C_2$), human households (HH), industry sector (IS), energy producer (EP), fuel source, water reservoir, resource pool, inaccessible resource pool.

species, the IS products, and the non-renewable energy source. $H_1$ has regulated grazing rights to access $P_2$. The access of $C_1$ to $H_1$ is regulated because while $H_1$ is private property, $C_1$ is a protected species. $H_2$ has limited access to $P_1$ by human action, e.g., fencing. There are four industries ($P_1$, $H_1$, IS, or EP) for the human workforce to work in, and IS determines appropriate wages based on the demand supply gap of the IS product along with the human population. The demand for any product ($P_1$, $H_1$, IS, and EP) also depends on the price and demand of other products. Some logical constraints are also modeled in such that with an increase in the price of a certain product, its demand decreases, and its demand increases with an increase in the price of other products. Typically, the wages paid for labor and the demand supply gap of a product determine the respective price of that product. Thus, it follows that as the wage levels increase or the demand supply gap increases, there is an increase in the price of a product. The price of energy depends on the labor and the amount of fuel that is available at that point in time. As the cost of labor increases, the price of energy also increases. Also, as the amount of available energy decreases, it causes a surge in the production and supply of energy to

minimize the demand supply gap. Some assumptions are modeled in this study, such that the price of energy generated from biopower is equal to the price of energy produced from the non-renewable energy source. The growth of the human population depends on the per capita mass, the birth rate, and the mortality rate. The mortality rate is a function of the crude death rate data for the past 65 years, starting from 1950. The human birthrate is assumed to be a negative function of the real wage rate as it represents the cost of opting to remain outside the labor force to raise children. The food-web is modeled by Lotka–Volterra type expressions, whereas the economy is represented by a price-setting model [14, 15] wherein the firms and HH attempt to maximize their well-being. The food-web model L-V expressions for the ecological compartments are:

***Primary Producers ($P_1$, $P_2$, $P_3$):***

$$\frac{dy_{P_1}}{dt} = y_{P1}\left(g_{RPP1}y_{RP} - g_{P1H1}y_{H1} - \frac{g_{P1H2}y_{H2}}{1+y_{HH}} - g_{P1HH}y_{HH} - m_{P1}\right) - g_{P1IS}y_{HH} \tag{1}$$

$$\frac{dy_{P2}}{dt} = y_{P2}\left(g_{RPP2}y_{RP} - \frac{g_{P2H1}y_{H1}}{1+y_{HH}} - g_{P2H2}y_{H2} - g_{P2H3}y_{H3} - m_{P2} + r_{IRPP2}y_{IRP}\right) \tag{2}$$

$$\frac{dy_{P3}}{dt} = y_{P3}(g_{RPP3}y_{RP} + r_{IRPP3}y_{IRP} - g_{P3H3}y_{H3} - m_{P3}) \tag{3}$$

***Herbivores ($H_1$, $H_2$, $H_3$):***

$$\frac{dy_{H_1}}{dt} = y_{H1}\left(g_{P1H1}y_{P1} + \frac{g_{P2H1}y_{P2}}{1+y_{HH}} - \frac{g_{H1C1}y_{C1}}{1+y_{HH}} - g_{H1HH}y_{HH} - m_{H1}\right) \tag{4}$$

$$\frac{dy_{H_2}}{dt} = y_{H2}\left(\frac{g_{P1H2}y_{P1}}{1+y_{HH}} + g_{P2H2}y_{P2} - g_{H2C1}y_{C1} - g_{H2C2}y_{C2} - m_{H2}\right) \tag{5}$$

$$\frac{dy_{H_3}}{dt} = y_{H3}(g_{P2H3}y_{P2} + g_{P3H3}y_{P3} - g_{H3C2}y_{C2} - m_{H3}) \tag{6}$$

***Carnivores ($C_1$, $C_2$, $C_3$):***

$$\frac{dy_{C_1}}{dt} = y_{C1}\left(\frac{g_{H1C1}y_{H1}}{1+y_{HH}} + g_{H2C1}y_{H2} - m_{C1}\right) \tag{7}$$

$$\frac{dy_{C_2}}{dt} = y_{C2}(g_{H2C2}y_{H2} + g_{H3C2}y_{H3} - m_{C2}) \tag{8}$$

***Humans (HH), Resource Pool (RP) and Inaccessible Resource Pool (IRP)***:

$$\frac{dy_{HH}}{dt} = y_{HH}(g_{P1HH}y_{P1} + g_{H1HH}y_{H1} - m_{HH}) \tag{9}$$

$$\frac{dy_{RP}}{dt} = m_{P1}y_{P1} + m_{P2}y_{P2} + m_{P3}y_{P3} + m_{H1}y_{H1} + m_{H2}y_{H2} + m_{H3}y_{H3}$$
$$+ m_{C1}y_{C1} + m_{C2}y_{C2} - y_{RP}(g_{RPP1}y_{P1} + g_{RPP2}y_{P2} + g_{RPP3}y_{P3} + g_{RPIRP})$$
$$- g_{RPIS}y_{IS} \tag{10}$$

$$\frac{dy_{IRP}}{dt} = (y_{IRP}(g_{ISIRP} + g_{ESIRP}) - y_{RP}(g_{RPP2}y_{P2} + g_{RPP3}y_{P3} + g_{IRPRP})) \tag{11}$$

It is to be noted that the term "$1 + y_{HH}$" represents human action where applicable. In case of (1), the access of $H_2$ to $P_1$ is regulated by humans so the corresponding growth term is a function of human population. Similarly, in (4) $H_1$ has regulated access to $P_2$ and the access of $C_1$ to $H_1$ is controlled since $H_1$ is private property. So, the corresponding growth rates are functions of human population. The investment in fencing, hunting or other regulatory activities also increases with a larger human population and is included in Eqs (1)–(7).

The expressions represent the rate of change in mass "$y_i$" of the various compartments "$i$". The terms "$g_{ij}$" represent the transfer of mass from compartment "$i$" to compartment "$j$" and the terms "$m_i$" represent the mortality rate of a compartment "$i$". A detailed description of these terms is presented in the nomenclature section.

The model economic strategy consists of four steps namely the determination of wage rates along with the prices and production levels of the various industries, the determination of demands by various parts of the ecosystem, the supply of the appropriate amount of material to each of the component of the ecosystem and the determination of the population growth. This process is illustrated in Fig 2. The expressions [12–15] describe the general equations for wage rate, current compartment stock, prices, and production targets, respectively. In equations [14, 15], the term "$a$" represents the fixed component of price and production whereas the terms "$b$" and "$c$" terms represent the variable components based on wages and current stock for a compartment.

$$W_t = a_{wage} - c_{wage}((IS_t^{def} + IS_t - \bar{IS}/(\lambda_{RP} + \theta_{P_1}) - d_{wage}N_{HH}^t \tag{12}$$

$$\Delta S_j^n = S_j - \bar{S}_j + S_{def_j}{}^n \tag{13}$$

$$P_j^n = a_j^{Price} + b_j^{Price}W^n - c_j^{Price}\Delta S_j^n \tag{14}$$

$$T_j^n = a_j^{Prod} - b_j^{Prod}W^n - c_j^{Prod}\Delta S_j^n \tag{15}$$

While increase in industrialization and automation has brought in an increased industrial production and energy consumption, it has also contributed to rampant pollution and the obliteration of species and resources. Thus, it is important to consider economic growth along with the growth of pollutants and emissions to observe global sustainability from a modeling perspective. To further study the impact of non-conventional energy sources on the environment and to observe effect of renewable energy on sustainability, it is pertinent to study the global emissions of greenhouse gases such as carbon dioxide and nitrous oxide.

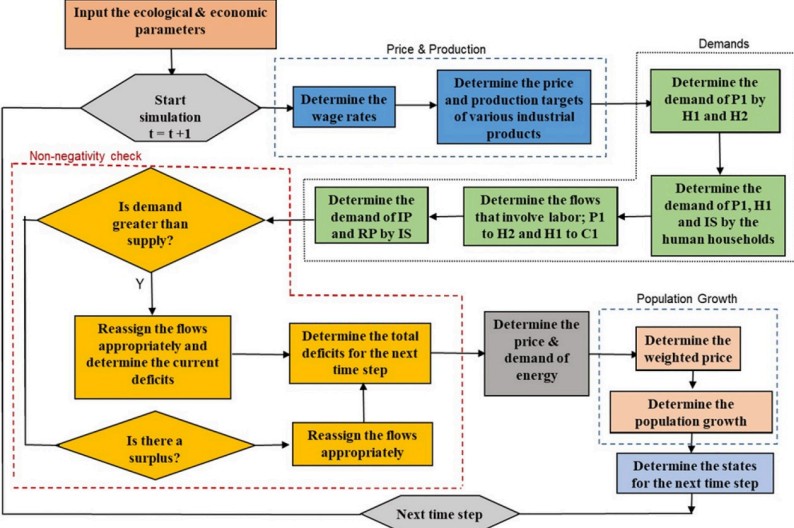

**Fig 2. Macro economic model.** Boxes with colors indicate- dark blue: step to determine wage rate, price of industry products and production targets. green: step to calculate demands of industrial products by various compartments. yellow: intermediate steps to supply appropriate amount of material to the compartments. orange: calculate population growth. Reproduced from [17].

These predictors can then be utilized to observe and accurately predict the influence of different strategies for global growth on the state of the world. The current model includes the capability to predict the carbon dioxide ($CO_2$) emissions, nitrous oxide ($NO_X$ or $N_2O$) emissions. and gross domestic product (GDP) to aid in the decision-making process. The general expressions for the carbon dioxide ($CO_2$) emissions, nitrous oxide ($NO_X$) emissions, and the global gross domestic product are derived by scaling the different compartments from the model to match the dynamic global data available from the year 1960 to 2020. The parameterization of these three variables was carried out using non-linear optimization techniques and appropriate historical data. The compartments contributing to the carbon emissions include $P_1$ and $H_1$ representing agriculture and livestock, IS representing industry, and EP representing energy sector. Additionally, the transportation sector, a major contributor to global carbon emissions, is considered as a function of the human population, and therefore the HH compartment. The same compartmental flows and the prices of the products associated with each of these compartments are used to compute the gross domestic product (GDP).

Nitrous oxide ($NO_X$) is a greenhouse gas and an atmospheric pollutant contributing to ecosystem-altering phenomena such as acid rain and overall global warming. The automotive, along with the agriculture and energy production sector, are well-known sources of $NO_X$ emissions. The impact of $NO_X$ emissions on the global ecosystem makes it a key element towards studying FEW sustainability. Thus, the flows from the different sectors namely, agriculture represented by $P_1$ and $P_2$, industry by IS, and energy sector by EP to the human sector, are used to model the $NO_X$ emissions.

Energy production utilizing fossil fuels has been one of the major contributors towards toxic emissions and pollutants in the modern world. Renewable energy has become a viable source of alternative energy in recent times through conscious, persistent efforts by scientists towards efficient energy harvesting and distribution measures. One such source of renewable energy in the form of "Biopower" is adopted as an alternative energy source in the current

model. Thus, the energy producer sector also utilizes the $P_1$ compartment when biopower is used. The amount of energy produced through biopower is flexible and can be varied.

The results from the validation of the model are presented in the next section.

## Model validation

In this section, the validation of the model is carried out by comparing the model results against available data. Time series data for the past 50-70 years were available for the human population, GDP, $CO_2$ emissions, and $NO_X$ emissions [18–21]. These data were used to validate the model. The validated model then becomes the foundation for the next part of the study and will be referred to in the following section as the "Base Case". The base case scenario represents the current state of the world. The base case scenario aims to understand and portray the dynamics of the global ecosystem. In the previous versions of this model, one deficiency was that the growth of human population was not explicitly represented. The current approach addresses this limitation by parameterizing the growth of the human population based on global data.

Owing to the global nature of the proposed model, historical data for the world is used to initialize the compartments wherever available. In this study, the population dynamics are modeled based on historical global mortality rate data available from the year 1960-2016 as opposed to the previous model where the human population change was not addressed [17]. The projections from the year 2020 onward are based on the data for mortality rate from UN world population insights 2019 report [21]. The phenomenon of sustainability needs to be studied over an extended period of time to observe broader effects on the ecosystem along with any short-term disturbances; thus, a long-time horizon of 200 years is selected to enable better understanding of these manifestations. The model is based on data that were available starting from the year 1950. Hence, the validation of the model compartments has been done from 1950-2019 and the model projections are for a 130-year timeframe from the year 2020-2150. Thus, the base case scenario depicts the growth of the global system from the year 1950 up till the year 2150. It can be observed that the human population predicted by GGSM from 1950-2020 shows a good fit against available data for global population as shown in Fig 3a. Therefore, the population predicted by the model is in congruence with global population for the past 70 years. The slight divergence from the year 2016 can be accounted to the fact that the real data for the mortality rate was available only till 2016. The legitimacy of the predictions for the population from GGSM was further investigated by comparing these predictions against population growth projections available from the UN world population insights study [21], as shown in Fig 3b.

It can be observed from Fig 3b that the growth of human population falls within the upper and lower variants and shows a good fit against the median variant of the population growth predicted by the United Nations report. Although GGSM predicts population growth which slightly deviates from the median variant for the 2020 to 2050 time period; it is still within 10% of the median variant from the UN model, thereby, justifying the population modeling approach in this model.

The agriculture sector ($P_1$) is initialized based on data for primary producers which was obtained from the GUMBO model [7]. Livestock in the world accounts for approximately 60% of the total biomass on Earth while, wild animals account for 4%, and humans contribute 36% [22]. These proportions are used to initialize the herbivore compartment $H_1$ and the carnivore compartment $C_1$.

The parameterization of $CO_2$ emissions, $NO_X$ emissions and GDP were carried out using non-linear optimization techniques by minimizing the squared error. Historical data for global

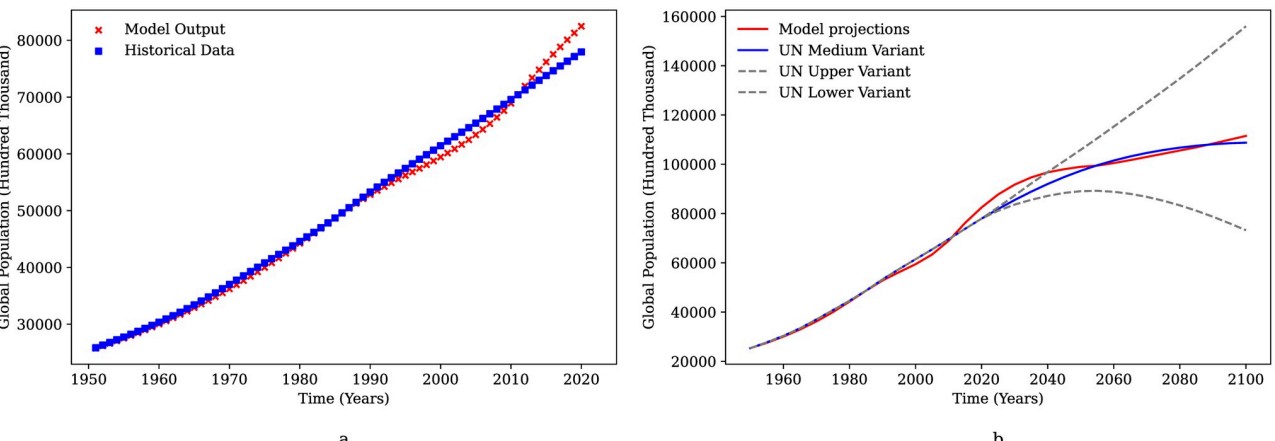

**Fig 3. Global population trends. a**: Comparison of the population predicted by GGSM denoted by "Model Output" to the historical global population. **b**: Human population from GGSM compared to projections for the global population growth from the UN world population insights report (1950-2100).

carbon emissions ($CO_2$) from the World Bank database for the 1960-2016 period was used to parameterize the $CO_2$ emissions from GGSM [18]. Fig 4a shows that the $CO_2$ emissions predicted by the GGSM model show a significant fit when compared to available data. Similarly, historical data for the global gross domestic product (GDP) available for the years 1960-2020 from the World Bank database [19] was used to parameterize the global GDP from the model. It can be observed from Fig 4b that the predicted GDP shows a close fit with global GDP.

As described in the previous section, agriculture, livestock, transportation, energy production, and industry sectors contribute to the total $CO_2$ emissions in the model. Thus, it is essential to attest the model reliability by comparing and verifying the sector wise emissions against available data to observe the influence of the individual sectors. The emissions from each of the above-mentioned sectors were compared to the data available for global sector wise emissions for the year 2009 to authenticate the sectoral accuracy of the model [23]. From the data

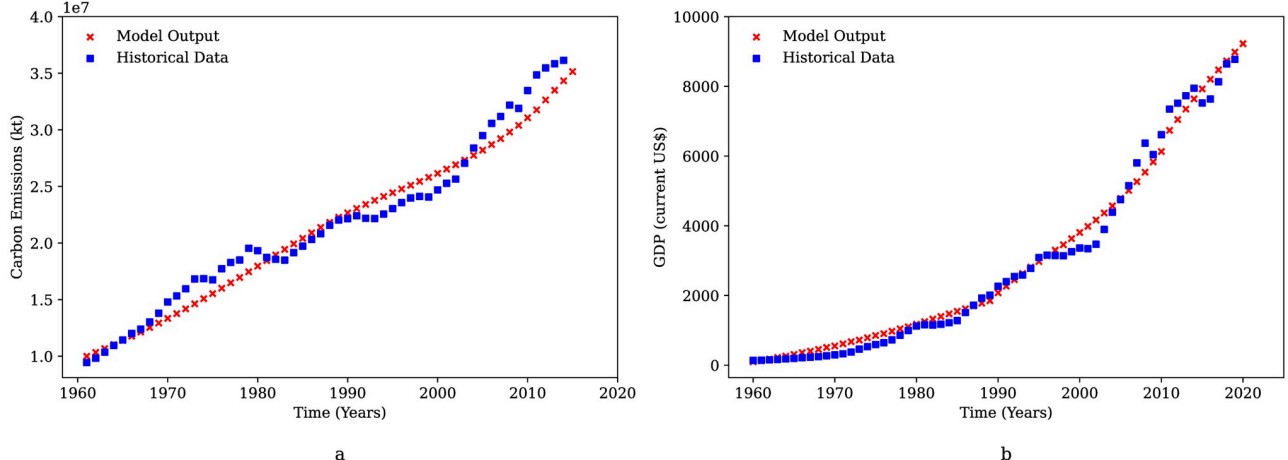

**Fig 4. Comparison of model predictions denoted as "Model Output" against available historical data: a: Carbon emissions ($CO_2$) b: GDP (USD).**

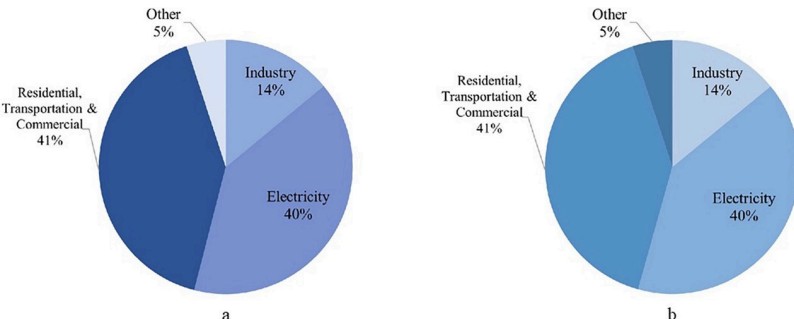

**Fig 5. Comparison of sector wise carbon dioxide ($CO_2$) emissions from the model to global sector wise carbon emissions for 2009 a: Data from UN energy statistics handbook (2009) b: GGSM model.**

for 2009, the electricity sector and the residential, transportation and commercial sector had the greatest influence on global carbon dioxide emissions as seen from Fig 5a. The same proportions can be observed from Fig 5b in the case of the current model. In the case of the model, the energy producer compartment represents the electricity component while the residential, transportation commercial component is represented as a function of the human population compartment in Fig 5b. This attests to the modeling approach for sectoral emissions in GGSM.

Nitrous Oxide emissions or $NO_X$ emissions computed by the model are compared against the available data for $NO_X$ emissions from the World Bank database for the 1970-2014 time period [20]. The fit of the nitrous oxide emissions from the model against the data can be observed in Fig 6. Even with the variability in available data, the $NO_X$ emissions from the model show a good fit against past data and can be used to provide a reasonable prediction for the future.

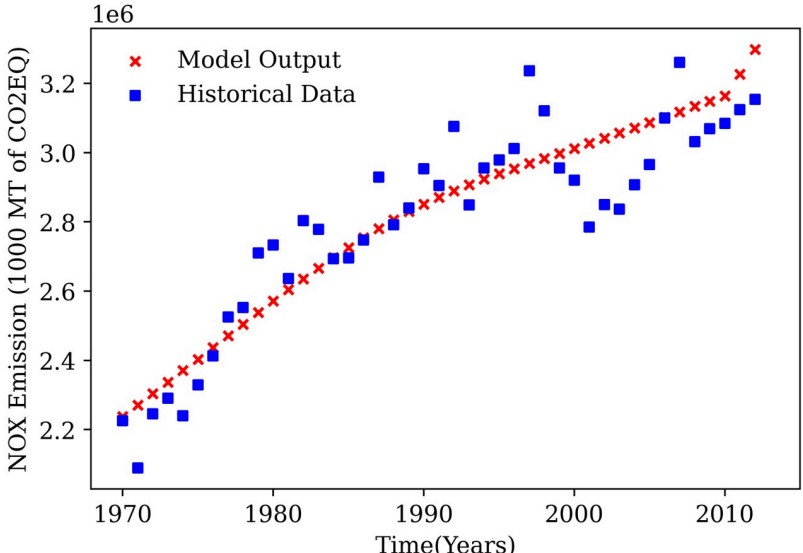

**Fig 6. Comparison of $NO_X$ emissions predicted by GGSM to historical global nitrous oxide emissions.**

This subsection showed the different parameters for which the generalized global sustainability model (GGSM) was validated for available data. This forms the base case for further analysis.

## Study of scenarios

Historically, scenario analysis has been conducted to study the intersecting sections of sustainability and to evaluate probable evolutionary pathways of the human-environment intersection [24]. The model is used to study different scenarios such as population explosion and increased resource consumption by humans. These possibilities are considered to pose a risk to global sustainability, and therefore, have been studied in the past [6, 16, 25–27].

The scenario in which the model parameters are best fit against historical data forms the base case for the model. The world population has been rising steadily for a long time. The development of better medical care, improved agricultural production, and better living conditions have resulted in a steadily decreasing mortality rate, resulting in population growth. However, as the human population increases, the stress on the finite natural resources of the Earth increases significantly. The world population has been increasing from 2.5 billion to 7.7 billion, at a rate of 1%—2% per year since 1950. More recently, increased focus on female education and access to medicine and contraception has led to decreasing global fertility rates. However, the world population is likely to still peak by the end of this century [28]. Hence, it is crucial to evaluate a scenario where the population grows at a much faster rate and to observe the effect of such rapid growth of the human population on the economy along with the ecological repercussions. In the current model, the population explosion scenario is modeled by utilizing the mortality rates for the upper variant of the population projected by the UN World Insights 2019 report.

As already mentioned, the availability of resources to support human activities is finite. The planet can only regenerate resources at a slow finite rate whereas humans have been known to consume the limited resources at an alarmingly high rate. A recent study states that the current consumption levels require an equivalent of 1.6 Earths for fulfilling the demand on Earth's ecosystem [29]. Some of these resources, such as energy sources, are non-renewable and limited, and the ecological resources which are renewable are being consumed at a faster rate than their regeneration rate allows. Environmentalists and climate scientists have been warning about the irreversible repercussions of such inordinate consumption. These consumption levels lead to a breakdown of ecosystems and put the delicate ecological balance of the world in jeopardy. However, the improved quality of living across the world, increased per-capita income, and technological developments have only added to the consumption levels especially in the energy sector where the demand is ever increasing. Previous studies have predicted that the consumption levels of most of the resources will increase by an average of about 50% over the next 50 years [6, 30]. In the present case, the increase in per capita consumption level is modeled by linearly varying the coefficients utilized for the estimation of the per capita demand of individual resources similar to previous work [17].

Thus, there are four scenarios namely the base case, population explosion, per capita consumption increase, and the combined scenario for population explosion with increased per capita consumption levels. These are referred to in the following section as scenarios 1—4, respectively. The profiles of different compartments for the above-mentioned scenarios starting from 2020 over a time frame of 130 years in the future to observe the system's stability and sustainability are presented below.

## Base case (scenario 1)

The results for all the compartments of GGSM for scenario 1 are presented in Fig 7. It can be observed that all the ecological compartments decrease slowly but steadily over time. The profile for $P_1$ from Fig 7 shows that primary producer $P_1$ stock decreases very slowly over time. This slow decrease can be assigned to the rising demand for $P_1$ due to the increasing population and limited availability of arable land and agricultural efficiency. The primary producer

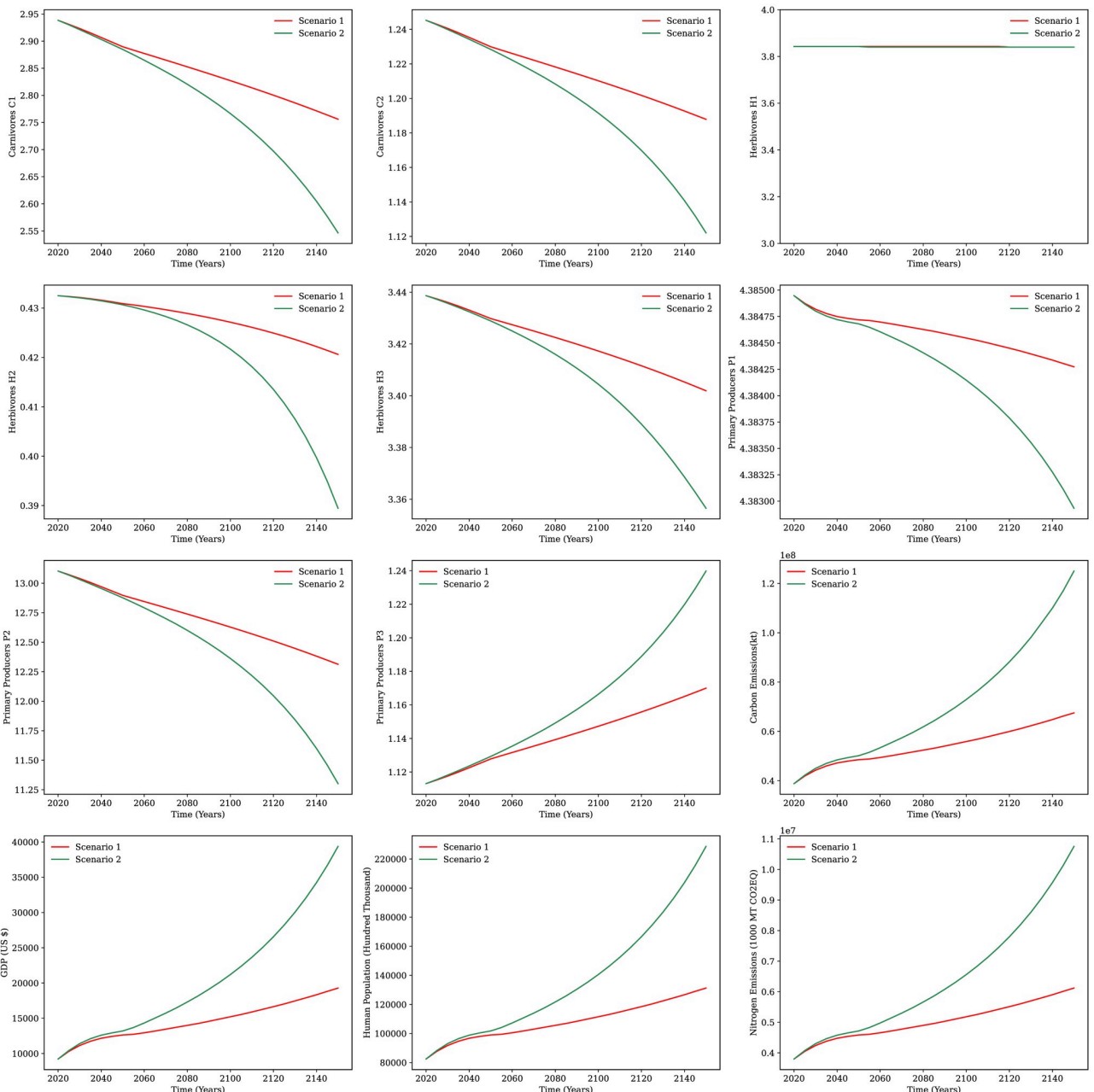

**Fig 7. Results for all compartments for the base case scenario (scenario 1) and the population explosion scenario (scenario 2).**

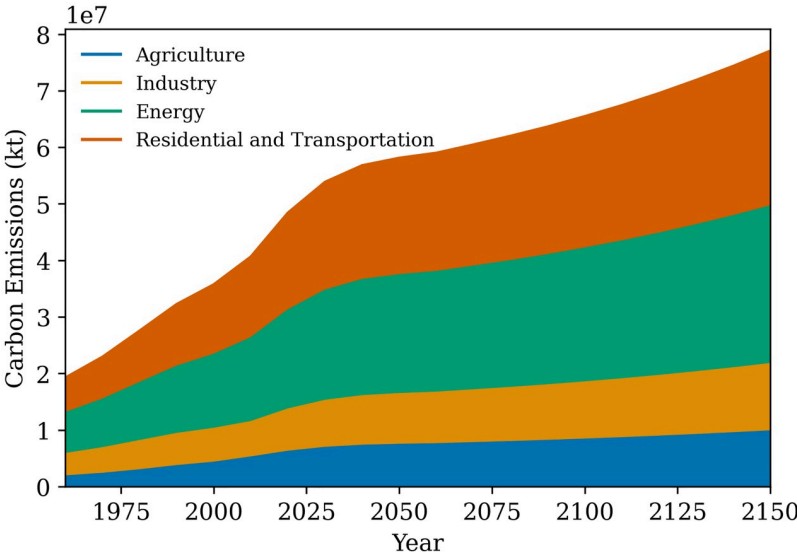

**Fig 8. Sector wise carbon emissions (2010-2150) for base case scenario.**

$P_2$ is utilized by $H_1$, $H_2$ and $H_3$ and hence decreases significantly over time. $P_3$ which represents wild species continues to grow at a slow rate. There is no significant change in the herbivore compartment of $H_1$. $H_1$ is a domesticated species and has economic value, so it is reasonable that the compartment mass is protected by the owners and does not change significantly over time. The changes in compartments $P_1$ and $P_2$ are also reflected in the $H_2$ and $H_3$ compartments which decrease over time. Thus, it can also be observed that the wild carnivorous animals $C_1$ and $C_2$ reduce slowly over time. GGSM predicts that the human population will grow at a steady rate for the next 130 years. Therefore, contributing to the increase in carbon emissions ($CO_2$ emissions) and the nitrogen emissions ($NO_X$ emissions). The base case results also show that the GDP will increase proportionately over the years.

The contribution of the different sectors towards the total carbon emissions ($CO_2$) for the base case is also presented in Fig 8. The energy producer sector and the residential and transportation sector are the largest contributors to the carbon emissions ($CO_2$). Thus, the results from the base case show that the system is stable. It can be said that if business as usual approach is utilized, the limited natural resources will slowly erode over time.

## Population explosion scenario (scenario 2)

As described previously, the upper variant of population projections from the UN World Insights 2019 report were used to model the population explosion scenario. United Nations projected the growth of the human population up until the year 2100 and the open-source data was used to determine how GGSM fared against other forecasts. Fig 9 shows that the human population forecasted by GGSM for the population explosion scenario is close to the upper variant projection of the global population by the UN model [21]. Fig 7 shows the results for scenario 2 as compared to the baseline results (scenario 1). For the population explosion scenario, all compartments show trends similar to the base case scenario. However, the ecological compartments for primary producers ($P_1$, $P_2$), herbivores ($H_2$, $H_3$) and carnivores ($C_1$, $C_2$) decrease steadily at a marginally faster rate compared to the base case. As the population increases rapidly, the carbon emissions ($CO_2$) and nitrous oxide ($NO_X$) emissions also increase

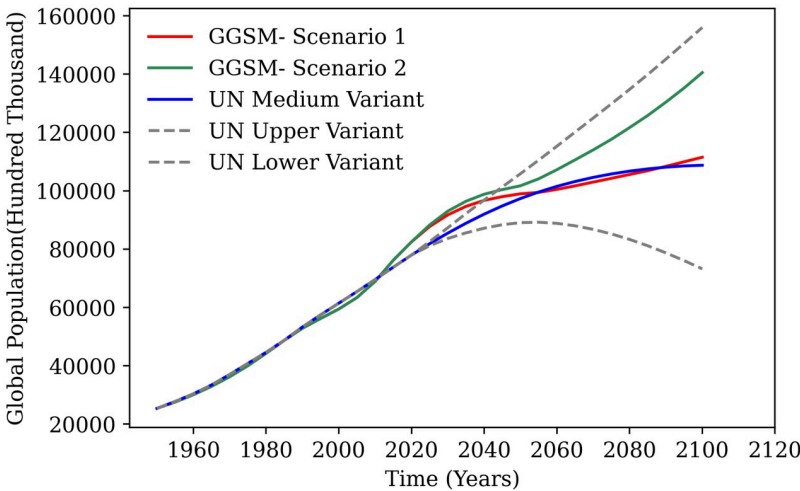

**Fig 9. Profile of human population compared to UN projections (1950-2100).**

significantly by a factor of 10 as opposed to scenario 1. The GDP also increases for scenario 2 with the increase in human population. The ecological compartment results show that the planet can support the increase in population if human consumption does not increase significantly.

## Per capita consumption increase scenario (scenario 3)

Fig 10 shows the behavior of all the compartments in GGSM when the consumption is increased in scenario 3. As the per capita consumption levels rise, there is a drastic impact on the ecological compartments. This is observed from the model from Fig 10 as species mass starts decreasing and revealing an uncertain and unstable ecosystem. The $P_2$ compartment declines first which results in the herbivores $H_1$ getting exhausted and leading to the rapid decline of wild animals represented by $C_1$ and $C_2$. The $P_1$ compartment holds up until a critical point, after which it declines rapidly as the consumption keeps on increasing. This decline can be attributed to climate change indicators such as soil erosion, exhaustion of symbiotic species, scarcity of water, along with unsustainable demand. As the per capita consumption levels increase, the carbon emissions ($CO_2$) and the nitrogen emissions ($NO_X$) increase by a large margin or by a factor of $10^2$ as opposed to scenario 1.

Since the GDP is a function of various other compartments, it is observed to be unstable in this scenario.

## Population explosion with per capita consumption increase scenario (scenario 4)

We know that in reality both global population and per capita consumption is increasing. This situation is evaluated by combining the population explosion scenario with the per capita consumption increase scenario.

It was observed previously in case of scenario 2 that the planet can support increased population if the per capita consumption levels do not increase. Also, it can be seen from the results for scenario 3 that increased consumption leads to steady exhaustion of resources. However, it can be said that increase in population cannot exist without increase in consumption. Thus, it

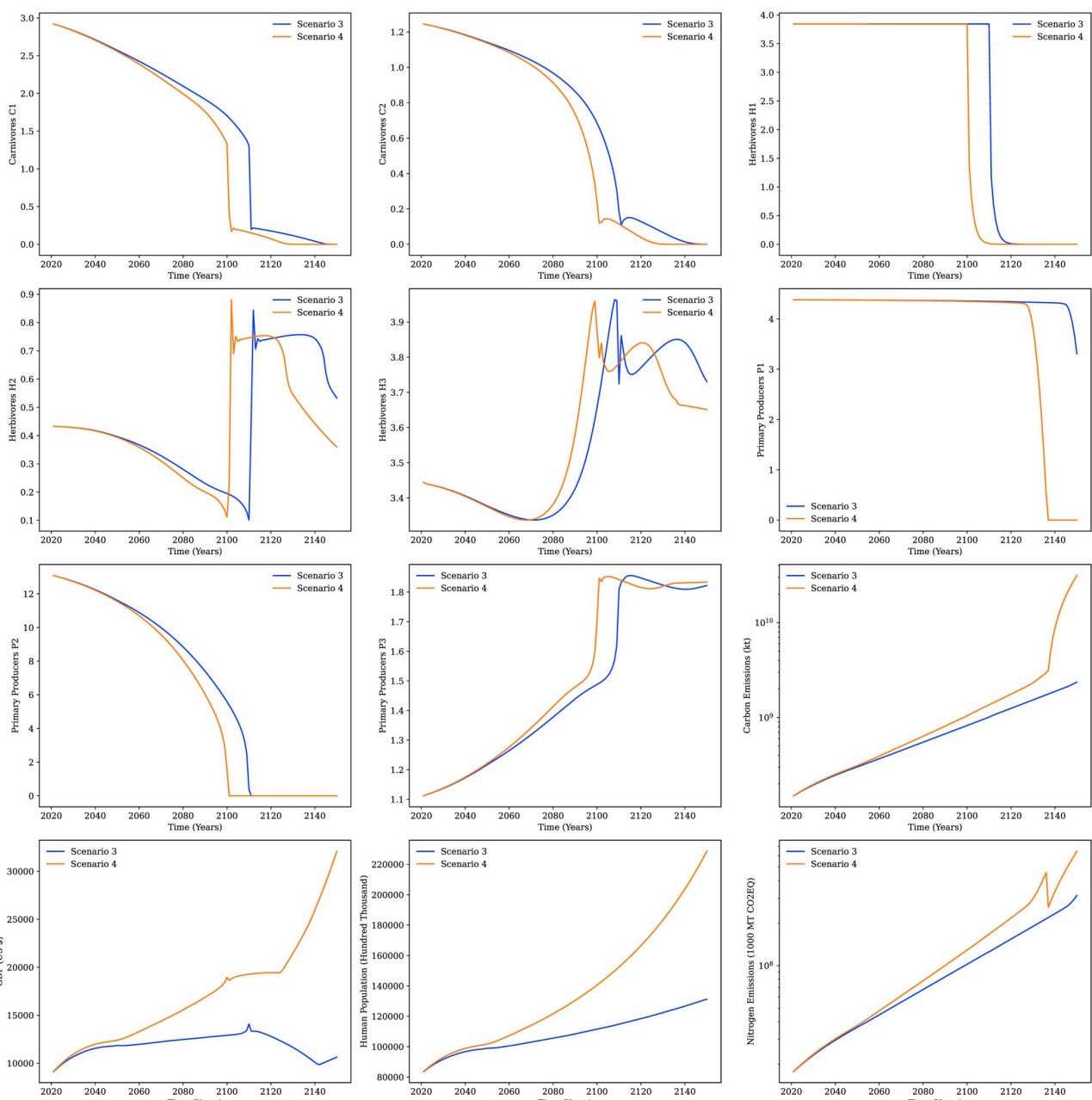

**Fig 10. Results for all compartments for the per capita consumption increase scenario (scenario 3) and population explosion with per capita consumption increase scenario (scenario 4).**

is imperative to study a scenario where the population increases, and so does human consumption. This scenario is presented here, and the results of this scenario are shown in Fig 10. The profiles for all the compartments for the population explosion combined with increased per capita consumption levels or scenario 4 compared to scenario 3 are presented in Fig 10. As the per capita consumption levels rise, so do the $CO_2$ emissions.

In case of scenario 4, the ecological compartment of $P_1$ starts getting depleted rapidly around the year 2130 as per capita consumption levels increase considerably. This decline and

subsequent exhaustion trigger a drastic increase in carbon emissions. As both the population and per capita consumption levels increase, the industry sector production increases and the system now consume conventional energy resources at a higher pace for energy production. The depletion of the $P_1$ compartment causes carbon emissions to increase drastically. This can be seen from Fig 10 for scenario 4, where the emissions increase significantly around the year 2135 as $P_1$ starts to deplete. Thus, the point after $P_1$ is exhausted, leads to an unstable system. After $P_1$ is exhausted, the carbon emissions increase significantly for scenario 4, where the population along with the consumption increases rapidly.

The $NO_X$ emissions for scenario 4, were observed to be much higher than the $NO_X$ emissions for scenario 1 by a factor of 100. It is observed from Fig 10 that the $NO_X$ emissions for the population explosion combined with the consumption increase scenario (scenario 4) are greater than those for the per capita consumption increase scenario (scenario 3). Thus, reasserting the fact that the system becomes unstable after the primary producer compartment $P_1$ is depleted.

It can be seen from scenario 4 trends that population explosion exacerbates the process as $P_1$ starts depleting and gets exhausted sooner. Thus, $P_1$ gets exhausted earlier for scenario 4 around the year 2135. As the per capita consumption levels increase GDP can be observed to be unstable and very sensitive to the system dynamics as observed from the results. Since GDP is dependent on many of the other compartments, a collapse in $P_1$ results in the collapse of the industry sector (IS) compartment which is the main contributing factor towards the peculiar pattern for GDP that is observed in Fig 10. The IS sector decreases sharply as the stores of $P_1$ are depleted and exhausted consequently and starts recovering slowly after the $P_1$ collapse. This process is exacerbated in case of scenario 4 and the timeline shifts left to show the depletion of $P_1$ and its effect on GDP.

The environment cannot support irrational consumption, but it can be controlled through global policy measures. Thus, policies targeted towards mitigating the impact of human consumption can be derived. As mentioned previously, GGSM can evaluate techno-socio-economic policies. An example is a policy that utilizes renewable energy in the form of biofuel-based power, or biopower can also be evaluated. Thus, policies will be derived using this model and evaluated further to control the impact of consumption increase and enhance sustainability.

## Conclusions

Changes in the current state of the world, along with climate change, have reiterated the necessity of a strategic and interdisciplinary study of sustainability. A generalized global sustainability model is presented here that aims at providing solutions and strategic policies to achieve sustainable development. To the best of our knowledge, the work presented here is the first-of-its-kind approach to study the sustainability across the food-energy-water (FEW) nexus on a global scale. One of the important contributions of this work is reparameterization of the model using historical data and validating the model results with the historical trends. Thus, the model is descriptive as well as prescriptive in nature and employs a granular approach. A base case scenario was presented to evaluate the model and establish its validity through comparison against data available for global parameters. The base case results show that the model is stable, and the predictions are in congruence with the global data available for the different compartments. It can be concluded that the planet could sustain an increased population if total consumption does not increase. As the human consumption levels increase the model results reiterate the fact that inordinate consumption by humans will heavily impact the global ecosystem and is unsustainable for future growth. It is also observed that increased population

along with increased consumption only exacerbates the process of resource depletion. This concludes the first part of GGSM work. The second part of this study includes modeling of the water compartment based on global water data and water demand and subsequent scenario analysis for the same.

## Future work

The analysis of several scenarios to evaluate sustainability with this model showed that indiscriminate human consumption would lead to a collapse of several resources in the future. An exponential increase in the human population exacerbates this process, with the resources getting exhausted sooner. Such circumstances warrant a study where sustainability is enhanced. Further, global policies need to be developed and evaluated to control several global conditions. An optimal-control theory-based approach will be studied in the future to evaluate sustainability by tracking and controlling multiple global indicators across different dimensions of sustainability. The Sustainable Systems Hypothesis proposed by H. Cabezas et al.(2018) requires maintaining at least six specific conditions necessary but not sufficient conditions for sustainable systems [31]. This hypothesis proposes that sustainability can be achieved by limiting the human burden on the environment so as to not exceed the biocapacity, conserving trophic and functional integrity of the ecosystem, ensuring adequate economic value production to exceed the value of consumption, maintaining certain quality of human existence, using energy resources in a sustainable manner and finally, maintaining system order and self-organization over time. These conditions will be captured through several sustainability indicators such as: Fisher information, green net product, ecological footprint, Greenhouse gas concentration in the atmosphere, and the global water stress [32]. Ecological footprint analysis (EFA) measures the equivalent land demand of the population by identifying the amount of bioproductive land required to support the annual average consumption and the waste production of an individual. Green Net Product (GNP) is a measure of sustainability from a macro-economic standpoint. GNP can be defined as the sum of all economic activities (all transactions) minus the loss value in the human-made and natural infrastructure. The economic performance of the society can be evaluated by the GDP, but sustainability needs the GNP. Greenhouse gas (GHG) concentration will be used to regulate the total CO2 eq contribution of the human sector activity and the mitigation by the natural sector. The optimal control model will study the multiple objectives defined above. The purpose of this study will be to look at the three crucial areas towards sustainability- ecological, economic, and social wellbeing. Thus, various policy options such as regulatory, economic, or technological parameters will be assessed as control variables, and a multivariate optimal control approach for global sustainability aimed at providing policy recommendations will be explored.

Prior work has studied the model as a network to determine the controllability of the system [33]. The study concluded that controlling a minimum of twelve nodes were necessary for sustainability in this system. These parameters have been used to conduct sensitivity analysis and are presented in Table 1. After conducting an initial sensitivity analysis, six variables have been narrowed down for the optimal control problem. These variables have the greatest impact on the ecological compartments and for maximizing sustainability. Some of the policy options that will be looked at include techno-social variables such as fines which penalize the Industry Sector (IS) and the energy producer sector (EP) for waste disposal. The socio-economic variables include the price of the compartment $P_1$ associated with the demand of $P_1$ by humans ($k_{P1HH}$) to evaluate plant consumption and similarly the price of the herbivores $H_1$ to evaluate the animal consumption ($p_{H1}$). The techno-economic variables include variables to evaluate the amount of $P_1$ required to produce a unit of the industrial product IS ($\theta_{P1}$), the grazing of

**Table 1. Policy options for sensitivity analysis.**

| Group | Name | Mass Flow | Significance of Variable | Selected |
|---|---|---|---|---|
| Techno-Social | Discharge fee ($d_{fee}$) | IS→IRP | Penalty imposed by governing body on the Industry sector(IS) for waste disposal. | Yes |
| Techno-Social | Energy production tax ($d_{EP}^{fee}$) | EP→IRP | Penalty imposed by governing body on the Energy Producer sector (EP) for waste disposal. | No |
| Techno-Social | Carbon tax ($t_{CO_2}$) | | Penalty imposed by governing body on the carbon emissions Currently being modeled | Yes |
| Socio-Economic | Plant consumption price ($k_{P1HH}$) | $P_1$→HH | Price of the compartment $P_1$ associated with the demand of $P_1$ by human households to evaluate plant consumption. | Yes |
| Socio-Economic | Animal consumption price ($p_{H1}$) | $H_1$→HH | Price of the compartment $H_1$ associated with the demand of $H_1$ by humans(HH) to evaluate animal consumption. | Yes |
| Socio-Economic | Agricultural price ($F_{P1H1}$) | $P_1$→$H_1$ | Price of the compartment $P_1$ associated with the demand of the demand of $P_1$ by $H_1$. | No |
| Socio-Economic | Plant material price ($p_{P1IS}$) | $P_1$→IS | Price of the compartment $P_1$ associated with the demand of by IS. | No |
| Socio-Economic | Industrial energy consumption price ($p_{EPIS}$) | EP→IS | Price of energy as associated with the demand of energy by IS. | No |
| Socio-Economic | Household energy consumption price ($O_{EPHH}$) | EP→HH | price of energy the demand of the demand of energy by Human Household. | No |
| Socio-Economic | Household Biofuel consumption price ($k_{EP}$) | EP→HH | price of EP associated with the demand of energy by Human household. | No |
| Socio-Economic | Price of water ($p_{WS}$) | | Price of water associated with the demand of water by Industry, Municipal or Agricultural Sector. Currently being modeled | Yes |
| Techno-Economic | Industrial agriculture demand ($\theta_{P1}$) | $P_1$→IS | Amount of $P_1$ required to produce a unit of industrial product IS. | Yes |
| Techno-Economic | Maximum grassland allowance ($\hat{k}$) | $P_2$→$H_1$ | Constant value specified by the government. Represents the grazing of $P_2$ by $H_1$. | Yes |
| Techno-Economic | Biopower Conversion rate conversion rate ($\lambda_{bio}$) | $P_1$→EP | Amount of $P_1$ needed to produce a unit of biopower. | Yes |

$P_2$ by $H_1$ ($\hat{k}$), and the amount of biomass ($P_1$) needed to produce a unit of biofuel ($\lambda_{bio}$). Work is being carried out to incorporate pricing of the water compartment. Further, a penalty will be imposed in the form of carbon tax for carbon emissions. The modeling of these variables is being carried out. Additionally, policies will also be derived based on these variables including, water price and carbon tax and others. These will be explored in-depth further.

The optimal control problem employing the six decision variables mentioned above and the objective to minimize the variance in the Fisher Information for the per capita consumption increase scenario (scenario 3) was solved as preliminary analysis. Fig 11 presents the comparison of all the compartments between uncontrolled scenario 3 and the optimal controlled scenario 3. These initial results show that the controlled scenario 3 is stable and prevents the collapse of the ecosystem while also reducing emissions and increasing GDP. However, the optimal control approach formulation comprises a big-scale optimization problem out of scope for the current study. A combination of several decision variables will be studied and controlled to regulate the unstable system. Then, multi-variable optimal control methods will be used to monitor the affected compartments and prevent exhaustion while maintaining sustainable growth. The methodology, sensitivity analysis, results, and comprehensive analysis of the optimal control strategy will be presented in a separate study in the future. The overarching objective of the study at this point is to demonstrate that sustainability is feasible and to illustrate the conditions that make that possible.

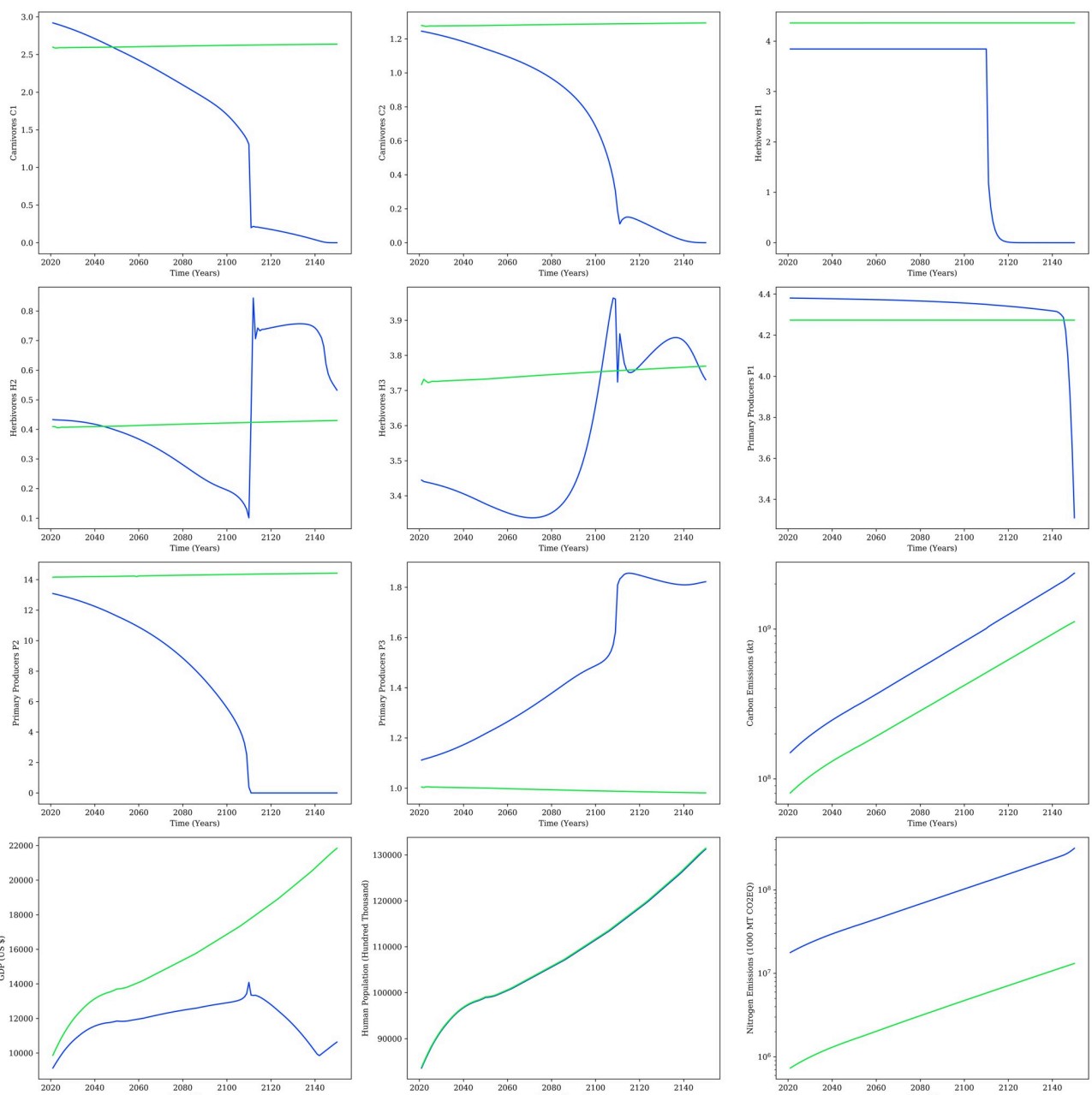

**Fig 11. Preliminary results.** Comparison of all compartments for the controlled per capita consumption increase scenario (scenario 3—controlled shown in green) to the uncontrolled per capita consumption increase scenario (scenario 3 shown in blue).

## Acknowledgments

This is a collaborative project between the USA, India, and Hungary. The authors would like to thank R. Boumans for sharing data from the Global Unified Model of the BiOsphere (GUMBO) and his invaluable inputs from the same.

## Author Contributions

**Conceptualization:** Urmila Diwekar.

**Formal analysis:** Urmila Diwekar.

**Investigation:** Apoorva Nisal, Urmila Diwekar, Neeraj Hanumante, Yogendra Shastri, Heriberto Cabezas.

**Methodology:** Urmila Diwekar.

**Project administration:** Urmila Diwekar.

**Software:** Apoorva Nisal.

**Supervision:** Urmila Diwekar.

**Writing – original draft:** Apoorva Nisal.

**Writing – review & editing:** Urmila Diwekar.

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
