## [Decision Letter · Decision Letter 0]

18 Jan 2022

PONE-D-21-28127Integrated model for food-energy-water (FEW) nexus to

study global sustainability: The main generalized

global sustainability model (GGSM)PLOS ONE

Dear Dr. Urmila Urmila,

Thank you for submitting your manuscript to PLOS ONE. After careful consideration, we feel that it has merit but does not fully meet PLOS ONE’s publication criteria as it currently stands. Therefore, we invite you to submit a revised version of the manuscript that addresses the points raised during the review process.

We look forward to receiving your revised manuscript.

Kind regards,

Indu K Murthy, Ph. D

Academic Editor

PLOS ONE

Journal Requirements:

https://journals.plos.org/plosone/s/file?id=ba62/PLOSOne_formatting_sample_title_authors_affiliations.pdf”

“This is a collaborative project between the USA, India, and Hungary. Collaborators 

from India acknowledge support from the Ministry of Human Resource Development, 

Government of India, through the SPARC (project code: P1238). The research 

contribution by H. Cabezas was carried out in the GINOP-2.3.2-15-2016-00010 

framework “Development of enhanced engineering methods with the aim at utilization 

of subterranean energy resources” project at the Research Institute of Applied Earth 

Sciences of the University of Miskolc, the Sz´echenyi 2020 Plan, partially funded by the 

European Union, co-financed by the European Structural and Investment Funds. The 

authors would like to thank R. Boumans for sharing data from the Global Unified 

Model of the BiOsphere (GUMBO) and his invaluable inputs from the same.”

“NH, YS, UD acknowledge the support from the Ministry of Human Resource Development,

Government of India, through the SPARC (project code: P1238).

The research contribution by H. Cabezas was carried out in the GINOP-2.3.2-15-2016-00010

framework Development of enhanced engineering methods with the aim at utilization

of subterranean energy resources" project at the Research Institute of Applied Earth

Sciences of the University of Miskolc, partially funded by the

European Union, co-financed by the European Structural and Investment Funds.

4. Please upload a copy of Figure 10, to which you refer in your text on page 17. If the figure is no longer to be included as part of the submission please remove all reference to it within the text.

Additional Editor Comments:

The manuscript is well written. However, there seems to be a disconnect in what the authors set out to do as indicated in the objective and the results presented. Further, some of the results presented are the obvious and do not really require analysis and scenario building. Request the authors to relook at the results presented. Given the study has adopted quite a robust methodology, it would be a good idea to revisit the study and come up with concrete, clear findings aligned with the objective.

Reviewers' comments:

Reviewer's Responses to Questions

**Comments to the Author**

1. Is the manuscript technically sound, and do the data support the conclusions?

Reviewer #1: Partly

2. Has the statistical analysis been performed appropriately and rigorously? 

Reviewer #1: N/A

3. Have the authors made all data underlying the findings in their manuscript fully available?

Reviewer #1: No

4. Is the manuscript presented in an intelligible fashion and written in standard English?

Reviewer #1: Yes

5. Review Comments to the Author

Reviewer #1: In the literature review section, the authors talk about how this model is different from other existing models in that it can be prescriptive and provide guidance on techno-socio-economic policies. However, in the article, there is no mention of these policies or outcomes from the modelling analysis. My suggestion to the authors would be to analyse the results towards being able to articulate policy suggestions (if that is the goal and USP of the model).

Regarding the scenarios, how is it possible to have population explosion without total consumption also increasing? Wouldn't scenario 2 and 4 be the same? If you mean per capita consumption, then that should be made clear.

The conclusion in this paper which is - increased consumption levels will pose stresses on natural resources and be unsustainable - is quite well-established. I would recommend that the authors spend more time to provide some unique results, with numbers and policy recommendations. The premise, objectives, methodology, and results/conclusions seem to be a bit disjointed. Perhaps this would be better as a methodology paper.

6. PLOS authors have the option to publish the peer review history of their article (what does this mean?). If published, this will include your full peer review and any attached files.

Reviewer #1: No

---

## [Author Response · Author response to Decision Letter 0]

7 Mar 2022

Integrated model for food-energy-water (FEW) nexus to study global sustainability: The main generalized global sustainability model (GGSM)

Apoorva Nisal1�, Urmila Diwekar1, 2�,

Neeraj Hanumante3‡, Yogendra Shastri3‡, Heriberto Cabezas4‡

1 Department of Industrial Engineering, University of Illinois, Chicago, IL, USA

2 Vishwamitra Research Institute, Crystal Lake, IL, USA

3 Department of Chemical Engineering, Indian Institute of Technology Bombay, Mumbai, Maharashtra, India

4 Research Institute for Applied Earth Sciences, University of Miskolc, Miskolc, Hungary

These authors contributed equally to this work.

‡These authors also contributed equally to this work.

* Corresponding author Email: 

urmila@vri-custom.org

Editor

Comment:

The manuscript is well written. However, there seems to be a disconnect in what the authors set out to do as indicated in the objective and the results presented. Further, some of the results presented are the obvious and do not really require analysis and scenario building. Request the authors to relook at the results presented. Given the study has adopted quite a robust methodology, it would be a good idea to revisit the study and come up with concrete, clear findings aligned with the objective.

Response: 

 Thank you for your invaluable comment and helpful suggestions. We have now modified the manuscript with additional information and a clear scope for future study to present a cohesive study. The methodology and modeling of the policy analysis using optimal control is a separate study in 

itself which is considerably different from the current endeavor and hence out of scope for the current manuscript. However, we have appended the manuscript with an additional section for future work where we present some preliminary findings from optimal control study and a scope for future study.

Reviewer 1

Comment: 

 In the literature review section, the authors talk about how this model is different from other existing models in that it can be prescriptive and provide guidance on techno-socio-economic policies. However, in the article, there is no mention of these policies or outcomes from the modelling analysis. My suggestion to the authors would be to analyse the results towards being able to articulate policy suggestions (if that is the goal and USP of the model).

Response:

Thank you for your valuable input and recommendations. To address these concerns, we have edited the manuscript and introduced certain policy suggestions and decision variables that we will be considering in the next part of this work. All of this is included in a separate future work section with a preliminary analysis.

This section is reproduced below.

Significant addition to previously submitted manuscript:

Future Work

The analysis of several scenarios to evaluate sustainability with this model showed that indiscriminate human consumption would lead to a collapse of several resources in the future. An exponential increase in the human population exacerbates this process, with the resources getting exhausted sooner. Such circumstances warrant a study where sustainability is enhanced. Further, global policies need to be developed and evaluated to control several global conditions. An optimal-control theory-based approach will be studied in the future to evaluate sustainability by tracking and controlling multiple global indicators across different dimensions of sustainability. The Sustainable Systems Hypothesis proposed by H.Cabezas et al.(2018) requires maintaining at least six specific conditions necessary but not sufficient conditions for sustainable systems [1]. This 

hypothesis proposes that sustainability can be achieved by limiting the human burden on the environment so as to not exceed the biocapacity, conserving trophic and functional integrity of the ecosystem, ensuring adequate economic value production to exceed the value of consumption, maintaining certain quality of human existence, using energy resources in a sustainable manner and finally, maintaining system order and self-organization over time. These conditions will be captured through several sustainability indicators such as: Fisher information, green net product, ecological footprint, Greenhouse gas concentration in the atmosphere, and the global water stress [3]. Ecological footprint analysis (EFA) measures the equivalent land demand of the population by identifying the amount of bioproductive land required to support the annual average consumption and the waste production of an individual. Green Net Product (GNP) is a measure of sustainability from a macro-economic standpoint. GNP can be defined as the sum of all economic activities (all transactions) minus the loss value in the human-made and natural infrastructure. The economic performance of the society can be evaluated by the GDP, but sustainability needs the GNP. Greenhouse gas (GHG) concentration will be used to regulate the total CO2 eq contribution of the human sector activity and the mitigation by the natural sector. The optimal control model will study the multiple objectives defined above. The purpose of this study will be to look at the three crucial areas towards sustainability- ecological, economic, and social wellbeing. Thus, various policy options such as regulatory, economic, or technological parameters will be assessed as control variables, and a multivariate optimal control approach for global sustainability aimed at providing policy recommendations will be explored.

Prior work has studied the model as a network to determine the controllability of the system [2]. The study concluded that controlling a minimum of twelve nodes were necessary for sustainability in this system. These parameters have been used to conduct sensitivity analysis and are presented in Table 1. After conducting an initial sensitivity analysis, six variables have been narrowed down for the optimal control problem. These variables have the greatest impact on the ecological compartments and for maximizing sustainability. Some of the policy options that will be looked at include techno-social variables such as fines which penalize the Industry Sector (IS) and the energy producer (EP) sector for waste disposal. The socio-economic variables include the price of the 

compartment P1 associated with the demand of P1 by humans (kP1HH) to evaluate plant consumption and similarly the price of the herbivores H1 to evaluate the animal consumption (pH1). The techno-economic variables include variables to evaluate the amount of P1 required to produce a unit of the industrial product IS (θP1), the grazing of P2 by H1 (kˆ), and the amount of biomass (P1) needed to produce a unit of biofuel

(λbio). Work is being carried out to incorporate pricing of the water compartment. Further, a penalty will be imposed in the form of carbon tax for carbon emissions. The modeling of these variables is being carried out. Additionally, policies will also be derived based on these variables including, water price and carbon tax and others.

These will be explored in-depth further.

The optimal control problem employing the six decision variables mentioned above and the objective to minimize the variance in the Fisher Information for the consumption increase scenario (scenario 3) was solved as preliminary analysis. Fig 11 presents the comparison of all the compartments between uncontrolled scenario 3 and the optimal controlled scenario 3. These initial results show that the controlled scenario 3 is stable and prevents the collapse of the ecosystem while also reducing emissions and increasing GDP. However, the optimal control approach formulation comprises a big-scale optimization problem out of scope for the current study. A combination of several decision variables will be studied and controlled to regulate the unstable system. Then, multi-variable optimal control methods will be used to monitor the affected compartments and prevent exhaustion while maintaining sustainable growth. The methodology, sensitivity analysis, results, and comprehensive analysis of the optimal control strategy will be presented in a separate study in the future. The overarching objective of the study at this point is to demonstrate that sustainability is feasible and to illustrate the conditions that make that possible. 

Fig 1. Preliminary results. Comparison of all compartments for the controlled consumption increase scenario (scenario 3 - controlled shown in green) to the uncontrolled consumption increase scenario (scenario 3 shown in blue)

Table 1. Policy options for sensitivity analysis

Group Name Mass Flow Significance of Variable Selected

Techno-Social Discharge fee (dfee) IS→IRP Penalty imposed by governing body

on the Industry sector(IS) for waste disposal. Yes

Techno-Social Energy production tax (dEPfee) EP→IRP Penalty imposed by governing body

on the Energy Producer sector (EP) for waste disposal. No

Techno-Social Carbon tax (tCO2 ) Penalty imposed by governing body on the carbon emissions

Currently being modeled Yes

Socio-Economic Plant consumption price (kP1HH) P1→HH Price of the compartment P1 associated with

the demand of P1 by human households to evaluate plant consumption. Yes

Socio-Economic Animal consumption price (pH1) H1→HH Price of the compartment H1 associated with

the demand of H1 by humans(HH) to evaluate animal consumption. Yes

Socio-Economic Agricultural price (FP1H1) P1→H1 Price of the compartment P1 associated with

the demand of the demand of P1 by H1. No

Socio-Economic Plant material price (pP1IS) P1→IS Price of the compartment P1 associated with

the demand of by IS. No

Socio-Economic Industrial energy consumption price (pEPIS) EP→IS Price of energy as associated with

the demand of energy by IS. No

Socio-Economic Household energy consumption price (OEPHH) EP→HH price of energy

the demand of the demand of energy by Human Household. No

Socio-Economic Household Biofuel consumption price (kEP) EP→HH price of EP associated with

the demand of energy by Human household. No

Socio-Economic Price of water (pWS) Price of water associated with

the demand of water by Industry, Municipal or Agricultural Sector.

Currently being modeled Yes

Techno-Economic Industrial agriculture demand (θP1) P1→IS Amount of P1 required to

produce a unit of industrial product IS. Yes

Techno-Economic Maximum grassland allowance (kˆ) P2→H1 Constant value specified by the government.

Represents the grazing of P2 by H1. Yes

Techno-Economic Biopower Conversion rate conversion rate (λbio) P1→EP Amount of P1 needed

to produce a unit of biopower. Yes

Comment: 

Regarding the scenarios, how is it possible to have population explosion without total consumption also increasing? Wouldn’t scenario 2 and 4 be the same? If you mean per capita consumption, then that should be made clear.

Response: 

The population explosion scenario (scenario 2) is modeled by modification of the human birth and mortality rates to capture the increase in population over the timeline. The per capita consumption is kept constant, so the total consumption increases. However, we also have another effect that is modeled in the system as population increases wage rate decreases. This is consistent across all scenarios. 

In case of the consumption increase scenario (scenario 3) the increase in the per capita consumption level of humans is modeled by linearly varying the constant coefficients involved in the estimation of per capita demand of resources. While in the case of the combined scenario (scenario 4), both the consumption and population increase together. This creates three different scenarios. We acknowledge that in a realistic population explosion scenario (scenario 2), the population cannot increase without an increase in consumption. However, this scenario is speculated to assert that the ecosystem can support population increase in isolation. Moreover, the environment cannot support irrational consumption, but it can be controlled through global policy measures. Thus, policies targeted towards mitigating the impact of human consumption can be derived. 

We have now clarified this in the revised manuscript.

Comment: 

 The conclusion in this paper which is - increased consumption levels will pose stresses on natural resources and be unsustainable - is quite well-established. I would recommend that the authors spend more time to provide some unique results, with numbers and policy recommendations. The premise, objectives, methodology, and results/conclusions seem to be a bit disjointed. Perhaps this would be better as a methodology paper.

Response: 

The idea behind this study is to present a novel global model based on historical data. Reparameterization of the model using historical data and validating the model results with the historical trends is one of the important contributions of this work. This sets the foundation for using the model for policy analysis. The model is validated based on historical data and evaluation of several sustainability scenarios are presented for analysis. This model can prescribe techno-socio-economic policies, but the evaluation of such policies is out of the scope for the current study. The validated model forms the basis of future work where such policies would be proposed and evaluated through an optimal control theory-based approach. Nevertheless, we have now added a future work section to give a prelude of work we plan on executing in the near future to assuage this particular concern from the reviewer. 

Additionally, we present preliminary results from a multi-variate optimal control approach to mitigate the exhaustion of species as observed in scenario 3. However, the formulation and solution of this optimal control approach includes a big optimization problem which is out of the scope for the current manuscript. Moreover, the results and analysis of policies derived from this strategy requires a separate study where they can be presented thoroughly. It should be kept in mind that the overarching purpose of this paper is an exposition of the methodology with some results to demonstrate the validity of the method and it use in policy formulation. An extensive study generating and analyzing policy recommendations will be the subject of a future paper.

References

1. Cabezas, H, et al. Design and engineering of sustainable process systems and supply chains by the P-graph framework Environmental Progress & Sustainable Energy. 2018; p. 624–636.

2. Benavides, P, et al. Controllability of complex networks for sustainable system dynamics Journal of Complex Networks. 2015; p. 566–583.

3. Rodriguez-Gonzalez, Pablo T and Rico-Martinez, Ramiro and Rico-Ramirez, Vicente s; 2020. Effect of feedback loops on the sustainability and resilience of human-ecosystem. Ecological Modelling. 2020; 426, 109018.

---

## [Decision Letter · Decision Letter 1]

8 Apr 2022

Integrated model for food-energy-water (FEW) nexus to

study global sustainability: The main generalized

global sustainability model (GGSM)

PONE-D-21-28127R1

Dear Dr. Urmila,

We’re pleased to inform you that your manuscript has been judged scientifically suitable for publication and will be formally accepted for publication once it meets all outstanding technical requirements.

Kind regards,

Ming Zhang, Ph.D.

Academic Editor

PLOS ONE

Additional Editor Comments (optional):

Reviewers' comments:

Reviewer's Responses to Questions

**Comments to the Author**

1. If the authors have adequately addressed your comments raised in a previous round of review and you feel that this manuscript is now acceptable for publication, you may indicate that here to bypass the “Comments to the Author” section, enter your conflict of interest statement in the “Confidential to Editor” section, and submit your "Accept" recommendation.

Reviewer #1: All comments have been addressed

2. Is the manuscript technically sound, and do the data support the conclusions?

Reviewer #1: Yes

3. Has the statistical analysis been performed appropriately and rigorously? 

Reviewer #1: I Don't Know

4. Have the authors made all data underlying the findings in their manuscript fully available?

Reviewer #1: Yes

5. Is the manuscript presented in an intelligible fashion and written in standard English?

Reviewer #1: Yes

6. Review Comments to the Author

Reviewer #1: (No Response)

7. PLOS authors have the option to publish the peer review history of their article (what does this mean?). If published, this will include your full peer review and any attached files.

Reviewer #1: No

---

## [Editor Report · Acceptance letter]

18 Apr 2022

PONE-D-21-28127R1 

Integrated model for food-energy-water (FEW) nexus to study global sustainability: The main generalized global sustainability model (GGSM) 

Dear Dr. Diwekar:

I'm pleased to inform you that your manuscript has been deemed suitable for publication in PLOS ONE. Congratulations! Your manuscript is now with our production department. 

Kind regards, 

on behalf of

Dr. Ming Zhang 

Academic Editor

PLOS ONE